# Single molecule kinetics of bacteriorhodopsin by HS-AFM

Alma P. Perrino [1], Atsushi Miyagi[1] & Simon Scheuring [1,2]✉

Bacteriorhodopsin is a seven-helix light-driven proton-pump that was structurally and functionally extensively studied. Despite a wealth of data, the single molecule kinetics of the reaction cycle remain unknown. Here, we use high-speed atomic force microscopy methods to characterize the single molecule kinetics of wild-type bR exposed to continuous light and short pulses. Monitoring bR conformational changes with millisecond temporal resolution, we determine that the cytoplasmic gate opens 2.9 ms after photon absorption, and stays open for proton capture for 13.2 ms. Surprisingly, a previously active protomer cannot be reactivated for another 37.6 ms, even under excess continuous light, giving a single molecule reaction cycle of ~20 s$^{-1}$. The reaction cycle slows at low light where the closed state is prolonged, and at basic or acidic pH where the open state is extended.

[1] Department of Anesthesiology, Weill Cornell Medicine, New York, NY, USA. [2] Department of Physiology and Biophysics, Weill Cornell Medicine, New York, NY, USA. ✉email: sis2019@med.cornell.edu

Bacteriorhodopsin (bR) is a membrane protein and member of the microbial rhodopsin family, a light-driving proton ($H^+$) pump formed by seven transmembrane helices (TMHs) and a covalently bound retinal chromophore[1]. Photon absorption induces the isomerization of the retinal from all-*trans* to 13-*cis*. This event starts a cascade of reactions that lead to $H^+$-release on the extracellular side and cytoplasmic $H^+$-uptake[2,3], thus $H^+$-pumping that establishes an electrochemical $H^+$-gradient that powers ATP-production.

The time scales at which the different light-induced intermediates of the reaction cycle occur, span from femtoseconds to milliseconds and have been studied by spectroscopy and structural methods[2–6]. We prefer to use the terminology "reaction cycle" that represents better the conformational and functional dynamics than the more spectroscopic term "photocycle". In brief, the bR reaction cycle can be subdivided into four main processes. First, upon photon absorption, the retinal isomerizes within the first picoseconds (ground to K state) and then relaxes in the 13-*cis* conformation (K and L state). Second, a $H^+$ from the retinal Schiff-base passes to Asp-85 (L to $M_1$ state) and the proton releasing group (Glu-194 and Glu-204) releases a $H^+$ to the extracellular side ($M_1$ to $M_2$ state). Third, through a large conformational change of helix-F a funnel to the cytoplasm opens and a $H^+$ enters via Asp-96 and re-protonates the retinal (N state). Fourth, the cytoplasmic gate closes (N to O state) and the protein resets for the next reaction cycle (O to ground state). The first two steps are accompanied by only small local conformational changes and happen within the first tens of microseconds. In contrast, the third step involves displacements of the cytoplasmic half of helices E and F and the E-F-loop (Fig. 1a, b), opening the cytoplasmic gate and rendering Asp-96 and the retinal accessible for re-protonation within the first ten milliseconds. In the fourth step, this conformational change is reversed and the cytoplasmic gate closes. Therefore, structurally, the reaction cycle can be divided into two major parts, the closed state, defined as the state where the cytoplasmic gate is closed (Fig. 1a, pink, PDB 6RNJ) and the open state (Fig. 1a, green, PDB 6RPH) where the E-F loop is displaced and the cytoplasmic gate is open, conceptually matching the alternate access model of membrane transport[7].

The functional aspects of the bR reaction cycle have been studied using spectroscopic methods[8–15]. In these experiments a large ensemble of bR is exposed to stimulating light pulses and the spectroscopic properties of bR and/or pH-sensitive labels is measured. According to these measurements, the K state and retinal isomerization occurs within picoseconds and the L state is reached after ~1.4 µs, a H+ is released on the extracellular side after ~630 µs, and the M state decays after ~4.7 ms. Finally, the decay of the N-state occurs after ~14 ms concomitant with re-protonation after ~12.1 ms[8]. This experiment could however not inform how long a prior active bR would need to reset from the O to the ground state and allow to be reactivated, and based on these measurements, it was assumed that bR cycles at ~100 s$^{-1}$.

The structural aspects of the bR reaction cycle have been studied by several techniques, some of them are also time-resolved methods: X-ray crystallography[4,16–18], solid-state NMR[19–21], cryo-EM crystallography[5,22–25], high speed atomic force microscopy (HS-AFM)[26–32] and, more recently, time-resolved serial femtosecond crystallography (TR-SFX) and X-ray free electron laser crystallography (XFELs)[3,7,33]. TR-SFX resolved the conformational changes in the retinal in the first femtoseconds[3,34]. XFEL and X-ray crystallography reported about secondary structure conformational changes in the later millisecond range after photoactivation[7,35]. Notably, a closed state with 13-*cis* retinal accumulated in the 0–5 ms-structure, while an open state with large displacement of the E-F loop accumulated in the 10–15 ms-structure with a decay of 23.8 ms. In these experiments the reverse conformational change to the O state could not be observed likely because the crystal contacts did not allow such rearrangements[7]. According to this work, the bR molecules complete the reaction cycle within 200 ms within the 3D-crystals, thus cycle at ~5 s$^{-1}$. Alike the spectroscopic analyses, X-ray and cryo-EM crystallography and NMR based methods provide ensemble averaged data.

Here, we develop and apply HS-AFM line-scanning[36,37] with millisecond temporal resolution coupled to a light activation system[26–28] to monitor single molecule structural dynamics of bR-WT and bR-D96N in native purple membranes. We find that large conformational changes to open the cytoplasmic gate occur within <3 ms upon light activation and persist for ~13 ms. Surprisingly, a prior activated molecule cannot be reactivated for another ~37 ms, and thus the maximum single molecules turnover rate is ~20 s$^{-1}$.

## Results

To measure single molecule kinetics of bR upon light activation, we integrated a green laser into our HS-AFM setup to provide 520-nm photons during defined duration and of defined intensity over ~15 µm diameter of the sample around the HS-AFM tip (Supplementary Fig. 1) while recording the protein dynamics (Fig. 1c). Similar to former AFM configurations[26–28,30,38], the infrared (IR) laser (HS-AFM laser) is reflected from the backside of the cantilever to measure its position and operate the HS-AFM feedback loop, while a second laser emitting light at 520 nm (activation laser) is used to stimulate the bR. A beam splitter in the optical path diverts part of the activation laser towards a light meter, which is monitored concomitantly to HS-AFM image acquisition. A signal generator connected to the activation laser driver allows to generate laser pulses of varying duration and intensity.

**Sidedness assignment of bR conformational changes**. In the TR-SMX structures upon light-activation[7] essentially no conformational changes are detected on the extracellular side of bR, while the cytoplasmic side displayed a closed state (Fig. 1a, b, pink, PDB 6RNJ) and an open state (Fig. 1a, b, green, PDB 6RPH), in which the cytoplasmic half of helix-F and the E-F loop move towards the periphery of the trimer. To assign the bR sidedness in our experiments, we first let purple membranes of bR-D96N adsorb at high density on the mica HS-AFM sample support, such as to find membranes exposing the extracellular and the cytoplasmic surfaces right next to each other. Second, we acquired HS-AFM movies at the interface of neighboring patches while applying green light (Fig. 1d). We reproducibly observed conformational changes only on one surface type of bR that we assigned to the cytoplasmic surface, in agreement with earlier bR sidedness assignments[39], while the extracellular surface remained unchanged (Supplementary Movie 1).

**Dynamics of bR-D96N**. Focusing on the reaction cycle-dependent conformational changes on the cytoplasmic bR side, we acquired high-resolution movies of bR-D96N using our laser combined HS-AFM (Fig. 1c). Because the retinal re-protonates from Asp-96, the mutation Asp-96-Asn (D96N) leads to slowing of the reaction cycle. Due to the slow inactivation of bR-D96N, we are able to observe both states at HS-AFM imaging scan rates (Fig. 1e, f). Applying single (Fig. 1e, Supplementary Movie 2) or multiple (Fig. 1f, Supplementary Movie 3) green laser sample illumination periods, the molecular response of bR-D96N was monitored and the average topography of the molecules in each frame calculated (Fig. 1e, f, middle). In addition, we measured the light-induced E-F loop displacement, the transition from the M to the N state, of each protomer in the movies by analyzing the peak position shift with respect to the trimer center. The average loop displacement of the protomers in the movies is ~5.0 ± 0.8 Å (Fig. 1e, f, bottom). Recently, we introduced localization AFM (LAFM[40], a method to

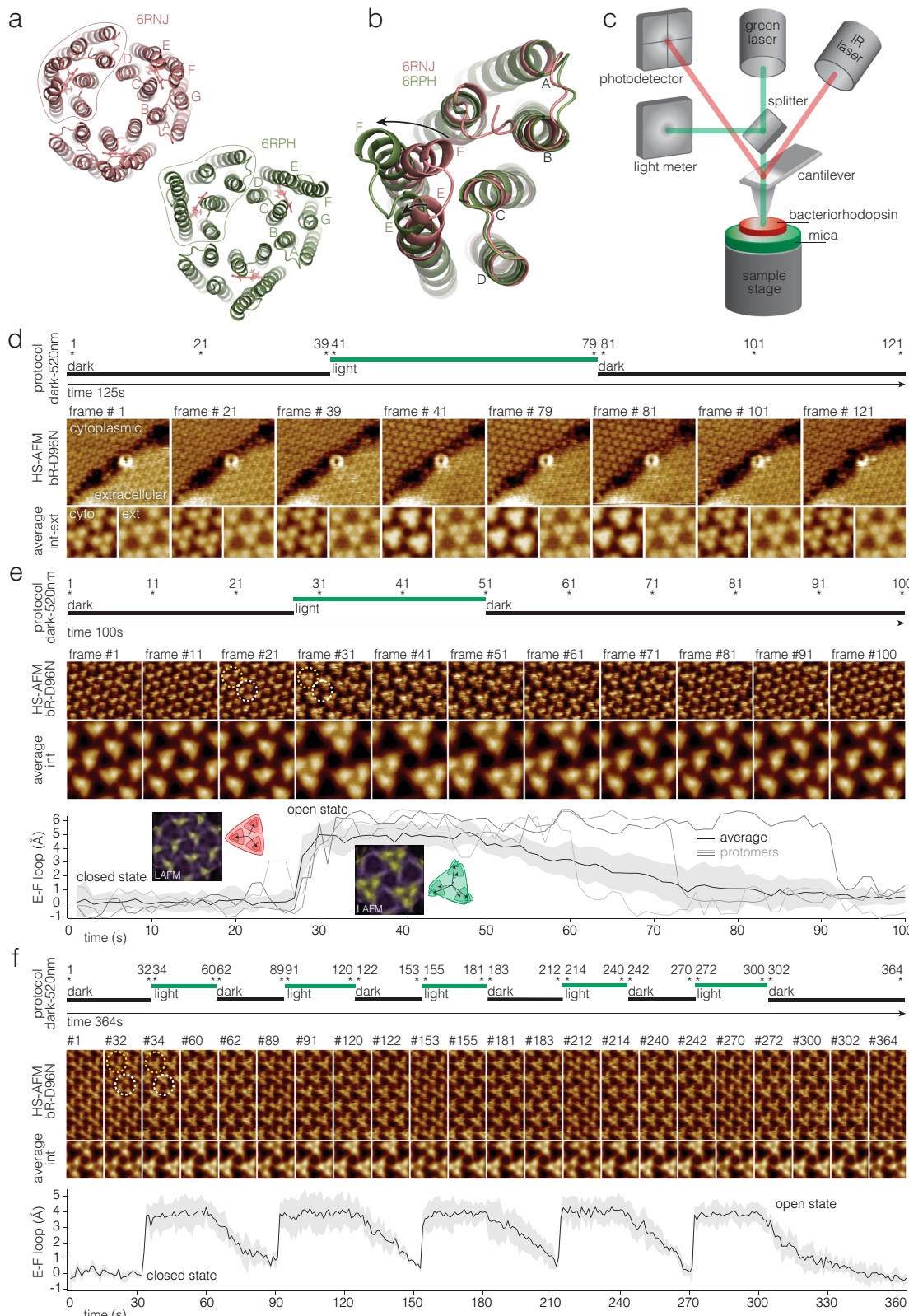

extract high-resolution information from AFM topographies: LAFM maps reveal an E-F loop displacement of 5.5 Å towards the periphery of the bR trimer with a counter-clockwise shift of 9° (Fig. 1e, bottom, insets). These measurements are in good agreement with former HS-AFM, cryo-EM and TR-SXM[7,26,41,42] data (see E-F loop displacement in Fig. 1b). Single protomer dynamics analysis provided additional information: while the average decay

time of the open state in bR-D96N after switching off the activation laser, the transition from the N to the O state, is ~10.5 s at pH 7, in good agreement with prior HS-AFM measurements of the D96N mutant[26–28,30,38], protomers display individual behavior and may reset swiftly or stay in the open state for ~40 s (Fig. 1e, bottom, gray lines). Another way to visualize the protomer behavior is through plotting the HS-AFM movie as kymographs of profiles across the

**Fig. 1 Laser combined high-speed atomic force microscopy (HS-AFM) for the study of photo-activated conformational changes in bacteriorhodopsin (bR). a** Structure of bR trimer viewed from the cytoplasmic side in the closed (PDB 6RNJ) and open (PDB 6RPH) states. Letters label the 7 helices. Outlines highlight one protomer. **b** Superposition of the open (green) and closed (red) structural states: The conformational change in helix-F (and to a minor degree in helix-E) are highlighted by arrows. The E-F loop moves by ~5 Å. **c** HS-AFM setup: Only the main components are shown and labeled: The IR laser (AFM laser) monitors the cantilever position. The green laser (activating laser) is used to excite bR. **d** Top: Light protocol. Middle: HS-AFM images of bR-D96N patches exposing cytoplasmic and extracellular sides (labeled). Bottom: Averages of the cytoplasmic and extracellular side topographies (labeled cyto and ext in the first pair of images). Only the cytoplasmic side displays measurable conformational changes during and after (bR-D96N has ~10.5 s open state dwell-time) light stimulation. Light-induced conformational changes in bR-D96N subjected to (**e**) a single, and (**f**) multiple green laser light periods. Top: Light protocol. Middle: High-resolution HS-AFM images and corresponding correlation averages. bR trimer (white dashed circle) and trefoil, defined as the three nearest-neighbor protomers that gather upon light-activated conformational changes (yellow dashed circle). Bottom: E-F loop displacement (average (black line) ±standard deviation (gray shaded area), $n = 12$). Gray lines in (**e**, bottom) show the behavior of three individual protomers. Insets in (**e**, bottom) are LAFM maps[40] of molecules in the closed (frames 1–26) and open (frames 28–49) state.

individual molecules (Supplementary Fig. 2). Recording the displacement of the E-F loop during repetitive ~30 s activating light - ~30 s dark cycles showed that all proteins reset each time, can be re-activated and no desensitization occurred under such conditions in bR-D96N (Fig. 1f).

**Single molecule dynamics of bR-WT**. Due to the much faster reaction cycle of wild-type bR (bR-WT; ~1000 times faster than bR-D96N), HS-AFM imaging cannot monitor the conformational changes associated with the reaction cycle[26,30]. When recording HS-AFM movies at 4 frames per second with a pixel sampling of 1.7 Angstrom per pixel, an average shift of the E-F-loop position and an increased topography standard deviation can be detected in the frames when light activation is on, but no bR-WT single molecule kinetics can be extracted from such imaging data (Supplementary Fig. 3, Supplementary Movie 4).

To characterize the much shorter decay of the open state of bR-WT as compared to bR-D96N (~13 ms vs ~10 s) a different set of experiments had to be designed. Recently, we introduced HS-AFM line scanning (HS-AFM-LS) combined with automated state analysis for the characterization of single molecule conformational changes[36,37]. In HS-AFM-LS the y-dimension (slow scan axis) of the image acquisition is disabled, and instead of recording images, the central x-scan line is repeatedly recorded with strongly increased temporal resolution. For example, instead of taking 2 images with 300 scan lines per second, we record 600 times the same scan line per second, thus reaching a temporal resolution of 1.6 ms. Stacking of this scan line as a function of time results in a kymograph that reports about the structural changes of the contoured molecules (Fig. 2c). First, we recorded HS-AFM-LS under continuous light conditions (Fig. 2a, top) and observed individual bR-WT protomers undergo tens of conformational changes per second (Fig. 2a, bottom, Supplementary Movie 5). Prior, we used HS-AFM-LS to analyze the single molecule kinetics of Glt$_{Ph}$, a member of the glutamate transporter family that works by the so-called elevator mechanism, in which the transport domain moves vertically through the membrane, and HS-AFM-LS recorded the height changes as a function of time[37]. Here, in the case of bR, we found that the lateral displacement of the E-F loop provided a great advantage in the kymographs (Fig. 2a, bottom), namely that the loop moved from one lateral position to another lateral position, leading to correlated loss of height signal in one position with gain of height signal in another position (red and green arrowheads in Fig. 2a, bottom; red and green traces in Fig. 2b). This provides a particularly strong signal when the scan line coincides with the directionality of the loop motion (Fig. 2a, inset). Subtraction of the two anticorrelated height signals allows the calculation of a height-time trace with excellent signal-to-noise ratio (yellow trace in Fig. 2b) that can be idealized for state dwell-time analysis (black trace in Fig. 2b). Thus, HS-AFM-LS records the entry into and exit from the N state by the displacement of the E-F loop with

millisecond temporal resolution, and thus under continuous light the turnover of the reaction cycle. The states assignment was performed fully computationally using the STaSI algorithm[37,43] (see Methods). From the idealized state assignment traces, we recovered the dwell-times of the bR-WT closed (Fig. 2d, left) and open (Fig. 2d, right) states, which were, at $1.9 \cdot 10^{18}$ photons/cm²·s, $\tau_{closed} = 107.8 \pm 0.8$ ms and $\tau_{open} = 18.9 \pm 0.3$ ms, respectively. The closed state comprises all states in which the E-F-loop is in the position as in the dark state, thus O, ground, K, L, and M states. The open state is defined as the state in which the E-F-loop is in the position opening the cytoplasmic gate, thus the N state.

Next, we reasoned that the bR-WT reaction cycle and associated conformational changes should be related to changing light intensity and set our green activation laser driver to a three step protocol, 2 s low light, 2 s high light, and 2 s dark. Expectedly, single molecules responded to low light with low activity, i.e., rare conformational changes to the open state, and to high light with high activity, and no conformational changes were detected in the absence of light (Fig. 2e). Thus, we started to record extended height-time traces at varying light intensities (Fig. 2f). Again, activity correlated with light intensity. The raw data and idealized height-time traces revealed to the eye that the open state dwell-times were always of similar length independent of the light intensity, while the closed state dwell-times got shorter and shorter with increasing light (Fig. 2f, bottom traces). This behavior is expected, once a retinal absorbs a photon and the bR-WT is activated, it enters and completes its reaction cycle in average $\tau_{open} = 13.2$ ms, the lifetime of the N state (Fig. 2g, bottom). This time is solely related to the molecular processes of proton pumping, and independent of the environmental light availability. In contrast, the lifetime of the closed state includes the time it takes for the next photon to be absorbed and start a new cycle and should therefore decrease with increasing light intensity. As expected, the dwell-times of the closed state decreased with increasing light intensity (Fig. 2g, top), until it reaches a plateau value at $\tau_{closed} = 40.5$ ms (Fig. 2g, top inset). Thus, the single molecule reaction cycle in the purple membrane is $\tau_{cycle} = 53.7$ ms (13.2 ms + 40.5 ms), and the maximum H⁺-pumping rate is $k_{H^+-pumping} = 1/(\tau_{cycle}) = 18.6 s^{-1}$. It is worth noting that prior spectroscopic work showed that the quantum efficiency of bR at pH 7 was 0.65, meaning that 65% of the molecules that absorb a photon actually activate[44,45]. However, we increased the photon availability by 1.5 orders of magnitude above saturation, but $\tau_{closed}$ did not further shorten (Fig. 2g), thus a bR-WT protomer cannot cycle faster than ~20 s⁻¹.

**Opening delay time of bR-WT and bR-D96N following short light pulses**. Having measured the open (N) and closed (sum of O, ground, K, L, and M) state single molecule lifetimes in continuous illumination experiments, we next aimed to determine the speed of the opening of bR-WT. The period upon a short light pulse, before the opening of the cytoplasmic gate in the N state,

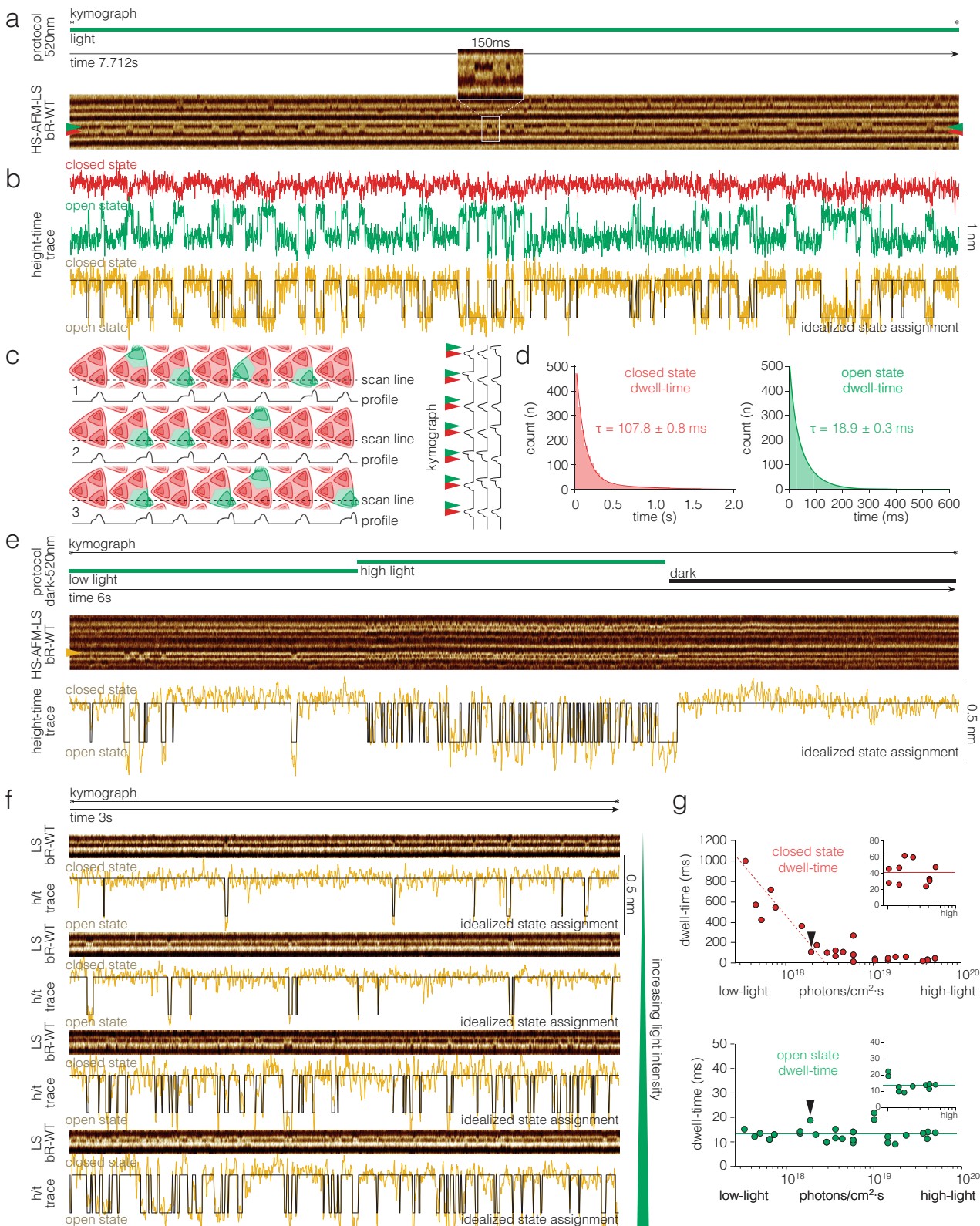

comprises states K, L, and M, and the period after the N state comprises O to ground state. Thus, this experiment allows us to split the closed state dwell into the two periods before and after the N-state. To measure this, we applied short activating laser-pulses to define the moment of photon capture to a precise time-point. We modulated the laser driver to produce one 0.5 ms (or 0.2 ms) short pulse every second (Fig. 3a, b, top), thus allowing us

to measure the time between photon capture by the all-*trans* retinal (ground to K state) and the conformational change in the E-F loop, i.e., the opening of the cytoplasmic gate that allows re-protonation of D96 (N state). From spectroscopic methods, we know that the all-*trans* to 13-*cis* transition is ultra-fast, within ~5 ps[3] - thus measuring the time that passes between photon-capture and the movement of the E-F loop corresponds to the

**Fig. 2 bR-WT single molecule kinetics of reaction cycle conformational changes under continuous light with millisecond temporal resolution. a** Top: Light protocol: Continuous light of predefined intensity. Bottom: HS-AFM-LS raw data kymograph. Red and green arrowheads indicate the closed and open state positions of a single protomer, respectively. Inset: 150 ms of the kymograph illustrating the clarity of the raw data comprising well resolved ~10 ms-events. The protomer E-F loop shifts position in the kymograph. **b** Height-time traces along the red (closed) and green (open) state E-F loop positions in **a**. To enhance the signal-to-noise ratio, we subtract the two anticorrelated signals (red and green traces) to get the yellow trace which is used for idealized state assignment (black trace, see Methods). **c** Schematic of bR trimers and the HS-AFM-LS x-scan line. The x-scan line contours closed (red) and open (green) protomers, giving location depend signatures in the contour profiles (below). The profiles are stacked to give a kymograph (right). Red and green arrowheads indicate the closed and open state positions as in **a**. **d** Dwell-time histograms of (**a**) and many more kymographs with closed state $\tau_{closed} = 107.8 \pm 0.8$ ms (red, $n = 483$) and open state $\tau_{open} = 18.9 \pm 0.3$ ms (green, $n = 471$) at $1.9 \cdot 10^{18}$ photons/cm$^2 \cdot$s light intensity. **e** Response of bR-WT protomers to changing light conditions. Top: Light protocol: Low light – high light – dark. Middle: HS-AFM-LS raw data kymograph. Yellow arrowhead: protomer whose activity is shown below. Bottom: Height-time trace (yellow) and idealized state assignment (black). **f** HS-AFM-LS raw data kymographs (top) and height-time traces (yellow) and idealized state assignments (black) (bottom) at different light intensities (the green triangle on the right symbolizes increasing light intensity from top to bottom). **g** Dwell-times in the closed (red, top) and the open (green, bottom) states as a function of light intensity. Error bars in **g** are smaller than the data points, see **d**. Black arrowheads in **g** indicate the data points from **d**. Insets: At saturating light power $<\tau>_{closed} = 40.5$ ms and $<\tau>_{open} = 13.2$ ms. $n$ is the total number of events analyzed. $\tau$ denotes dwell-time as the fit of the single exponential fit, $y(x) = Ce^{-x/\tau}$.

conformational coupling between the change in the retinal to the end of helix F. A recent TR-SFX study found this conformational change in the 10–15 ms post laser activation structure[7]. In our experiment, we can directly measure two observables on the single molecule level: 1) the time after photon burst to propagate the conformational change from the retinal to the opening of the cytoplasmic gate, which we term the opening delay time; and 2) the time the protein stays in the open state. The latter should be the same as the open state dwell-time determined in the continuous light experiments (Fig. 2).

To precisely synchronize the activation laser pulses with respect to the HS-AFM-LS recordings, we recorded the photodiode signal in a second data acquisition channel (see Methods). This allows us to precisely calculate the opening delay time from the first pixel when the green laser hits the sample to the pixel where the conformational change of a protomer is identified (Fig. 3a–d, Supplementary Fig. 4). In the presented experiments, scan lines are acquired every 1.667 ms and each pixel is recorded during 2.7 μs (Fig. 3c, d, Supplementary Fig. 4, see Methods). We applied this method to both, bR-WT (Fig. 3a) and bR-D96N (Fig. 3b). Due to the shortness of the activation laser pulses (Fig. 3a, b, top), only few protomers activated upon each pulse (Fig. 3a, b, bottom, dashed outlines). Close inspection of the conformational response of individual protomers showed that the E-F loop moved within the same or in one of the subsequent scan lines upon a green laser pulse, in both, bR-WT (Fig. 3c, left) and bR-D96N (Fig. 3d, left). Plotting the opening event times (with a bin width of 1.667 ms, corresponding to the time of a full scan line cycle), we derived the E-F loop conformational delay upon photo-activation through exponential decay fitting as $\tau_{delay} = 2.9 \pm 0.1$ ms ($n = 110$) for bR-WT (Fig. 3c, right) and $\tau_{delay} = 3.2 \pm 0.1$ ms ($n = 67$) for bR-D96N (Fig. 3d, right), respectively. We also applied a shorter laser pulse of 0.2 ms for bR-WT and obtained the same opening delay time, $\tau_{delay\text{-}0.2ms\text{-}pulse} = 2.9 \pm 0.1$ ms, but from lower protomer activation numbers ($n = 100$) (Fig. 3e, gray arrowhead). Comparing the opening delay times, bR-D96N was 10% slower than bR-WT, suggesting that not only re-protonation of the retinal is slowed down when aspartate-96 is substituted with an asparagine but also the conformational change to open the cytoplasmic gate was slightly slower; as has been observed for other bR-mutants[27]. The main impact of the D96N mutation is the delay in the re-protonation step. Indeed, the kymographs upon pulsed activation revealed that the bR-WT open state dwell-time, corresponding to the duration until the E-F loop regains its initial position, had a time constant $\tau_{open\text{-}WT} = 14.2$ ms (Fig. 3c, left, Fig. 3f, g), very similar to $<\tau>_{open\text{-}WT} = 13.2$ ms from the continuous light experiment (Fig. 2g). In contrast, the bR-D96N protomers remained open over extended kymograph acquisition (Fig. 3d, left; Fig. 3b,

Supplementary Fig. 5). The bR-D96N open state dwell-time is better measured using HS-AFM imaging, $\tau_{open\text{-}D96N} = ~10.5$ s (see Fig. 1a, b;[26]). Our experiment with short light pulses addresses similar questions from a conformational perspective as experiments that analyzed the spectroscopic response of bR and pyranine pH-probes following short laser flashes[8]. In these experiments, the rise of the N state occurred ~3.5 ms after flash. The pyranine pH-probe fluorescence decayed ~12 ms and the N state decayed ~14 ms after the laser flash. These results are in good agreement with $\tau_{delay} = 2.9$ ms and $\tau_{open} = 13.2$ ms, found here. In the TR-SMX study the closed state decayed after $4.7 \pm 0.3$ ms and the cytoplasmic gate open state structure peaked in the 10 ms to 15 ms bin and decayed after $23.8 \pm 2.7$ ms[7]. Thus, our conformational measurements are in good agreement with the spectroscopic data, while the conformational changes in the 3D-crystals appear somewhat slowed.

All the above described measurements were performed at neutral pH 7. However, given that bR is a H$^+$-pump, and that the inactivation step implicates re-protonation of D96, we next examined the opening delay times and open state dwell-times of the bR-WT at different pH (pH 5, pH 6, pH 7, pH 8 and pH 9). Interestingly, the opening delay time remains essentially unchanged over the entire pH-range, with an average value $<\tau>_{delay} = 2.7 \pm 0.3$ ms (Fig. 3e), meaning that the initial steps of the reaction cycle, photo-activation of the retinal, proton release on the extracellular face, and the conformational change along helix-F to the E-F loop, were pH-independent. In contrast, the open state is pH-dependent (Fig. 3f, g): At acidic and neutral pH 5, pH 6 and pH 7, a similar short open dwell with an average time constant $<\tau>_{open(pH5\text{-}pH7)} = 14.1 \pm 1.3$ ms is found, while at alkaline pH 8 and pH 9 $<\tau>_{open(pH8\text{-}pH9)} = 34.5 \pm 2.1$ ms is measured (Fig. 3h). It is quite intuitive that at acidic and neutral pH, the abundance of H$^+$ for re-protonation of the Schiff base leads to shorter open state dwell-times than at alkaline pH. However, interestingly, the dwell-time distributions show that the fastest and most uniform time constant is found at pH 7, while at lower pH, pH 6 and pH 5 the distributions have long tails (Fig. 3f). Fitting the low pH open state dwell-time distributions with two exponentials, we identify $\tau_{open\text{-}2(pH5)} = 82.2 \pm 4.7$ ms and $\tau_{open\text{-}2(pH6)} = 51.2 \pm 6.1$ ms (Fig. 3f, Supplementary Fig. 6). From the fit statistics, we estimate that ~10% of the molecules at low pH are in a slower photocycle with an extended open state dwell. It has been reported that at pH below 6 the proton release to the extracellular medium may directly be accomplished by Asp-85 instead of the proton releasing group (Glu-194 and Glu-204), which may lead to a slower photocycle[46]. While the single molecule kinetics show that the individual molecules can inactivate fast (~13 ms, 90%) or slow (~60 ms, 10%), in ensemble

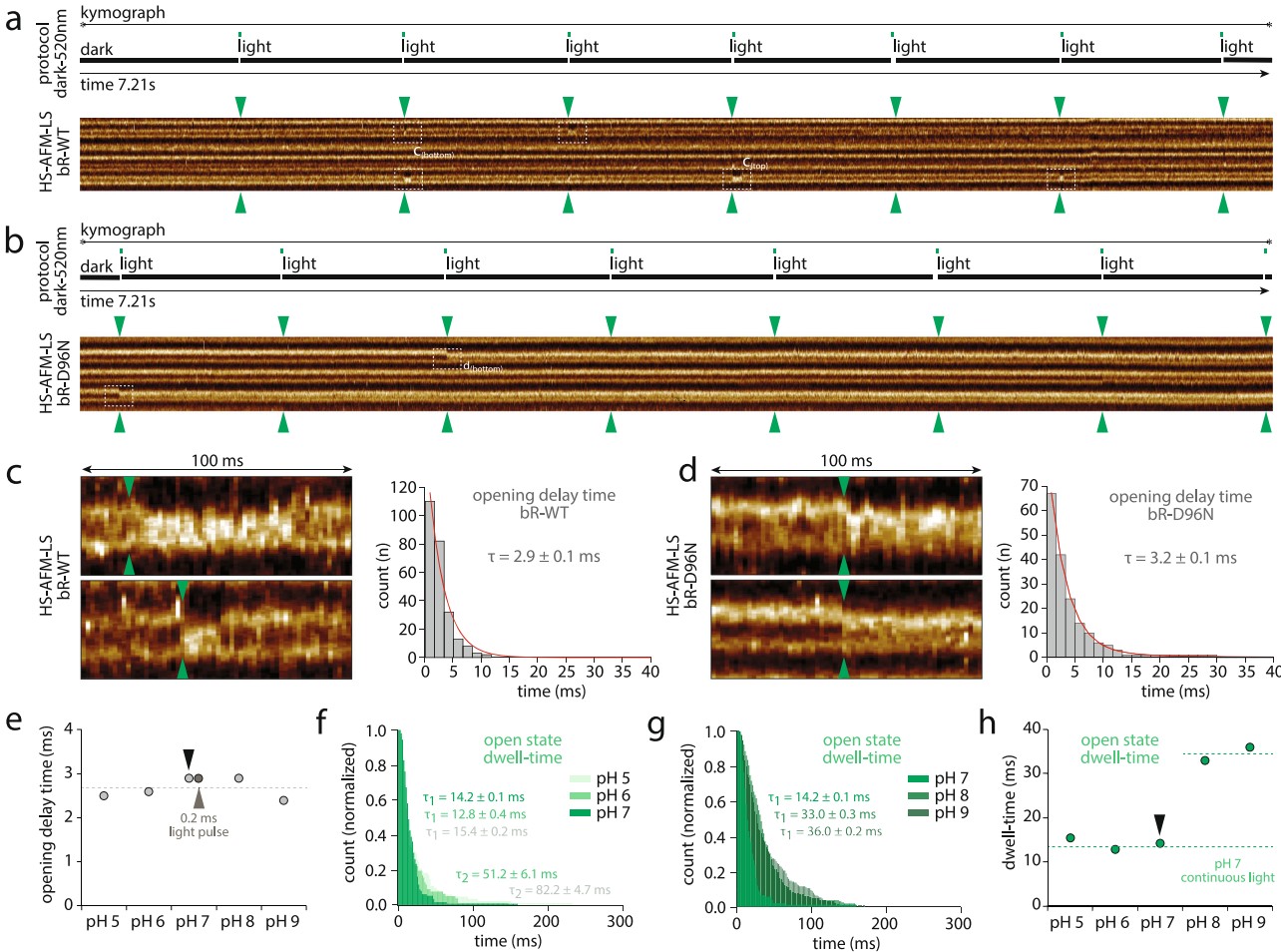

**Fig. 3 Time delay between photoactivation and opening of the cytoplasmic gate.** Top: Light protocol, and bottom: kymographs of (**a**) bR-WT, and (**b**) bR-D96N, in response to short activation light pulses (one 0.5 ms-pulse every second). The green arrowheads indicate the time of the activation light pulses. The white dashed outlines highlight bR protomer opening. Left: Raw data display of 8 nm scan dimension and 100 ms kymograph of the conformational changes occurring in individual (**c**) bR-WT, and (**d**) bR-D96N, protomers (see white dashed outlines in **a** and **b**). Right: Opening delay time distribution for (**c**) bR-WT with $\tau_{\text{delay-WT}} = 2.9 \pm 0.1$ ms ($n = 110$), and (**d**) bR-D96N with $\tau_{\text{delay-D96N}} = 3.2 \pm 0.1$ ms ($n = 67$). **e** $\tau_{\text{delay}}$ of bR-WT at pH 5, pH 6, pH 7, pH 8 and pH 9. Black arrowhead indicates the data point from **c**. Gray arrowhead indicates the data point from experiments with 0.2 ms-pulses. **f** Dwell-times of the open state at pH 5, pH 6, and pH 7. First component exponential fittings give open state dwell times of $\tau_{\text{open-1(pH5,pH6,pH7)}} = \sim 14$ ms (as indicated). Second exponential fittings for pH5 and pH 6 give $\tau_{\text{open-2(pH5)}} = 82.2 \pm 4.7$ ms and $\tau_{\text{open-2(pH6)}} = 51.2 \pm 6.1$ ms. **g** Dwell-times of the open state at pH 7, pH 8 and pH 9 (as indicated). **h** $\tau_{\text{open-1}}$ of bR-WT at pH 5, pH 6, pH 7, pH 8 and pH 9. Dashed line (bottom) indicates the open state dwell time obtained at pH 7 under continuous light conditions (see Fig. 2g, bottom). Dashed line (top) indicated the average value at pH 8 and pH 9. Note that at acidic pH 5 and pH 6, ~10% of the molecules adopt a long activate state with $\tau_{\text{open-2(pH5,pH6)}} = \sim 60$ ms. $n$ is the total number of events analyzed. $\tau$ denotes dwell-time as the fit of the single exponential fit, $y(x) = Ce^{-x/\tau}$, for pH > 7, and double exponential fits, $y(x) = C_1 e^{-x/\tau_1} + C_2 e^{-x/\tau_2}$, for pH 5 and 6.

experiments an extended average open state should be found at acidic pH, ~35% slower, and therefore an overall slower $H^+$-pumping in all conditions different from neutral pH. Thus, bR performs best at neutral pH 7, because $H^+$-release and re-protonation are most efficient under such conditions[46,47].

Most important, being able to define the rise dwell time, 2.9 ms, between photon capture and opening of the cytoplasmic gate (N state) from the flash activation experiments, allows us to unambiguously assign the dwell time in the closed state minus the rise time, 40.5 ms − 2.9 ms = 37.6 ms, to the transition from the O back to the ground state (see Discussion).

## Discussion

Since the discovery of the purple membrane and the identification of bacteriorhodopsin (bR) in *Halobacterium salinarum* (formerly *H. halobium*)[1] various techniques have provided deep insights about its structure and the $H^+$-pumping mechanism. Traditionally, the photocycle has been studied using spectroscopic methods[13–15] and

more recent time-resolved crystallography studies revealed the conformational changes from the femtosecond to the millisecond regime[3,7,33].

Here, we applied HS-AFM-LS in a laser-coupled HS-AFM system to characterize the single molecule kinetics of bR. Our study resolves the single molecule rate constants of the conformational changes at millisecond temporal resolution under different light intensities and at different pH: The reaction cycle is optimized to work at high light intensity and neutral pH. In short light pulse activation experiments, we found that the conformational change opening the cytoplasmic gate in bR-WT in the native purple membrane occurred within <3 ms upon photon-capture, and the open state was ~13 ms long. In continuous saturating light experiments, the open state was also ~13 ms long independent of the light intensity, and the closed state shortened with increasing light but could not get shorter than ~40 ms. Combining the two experiments (Fig. 4), we can calculate the dwell time to reset the molecule after closing the cytoplasmic gate

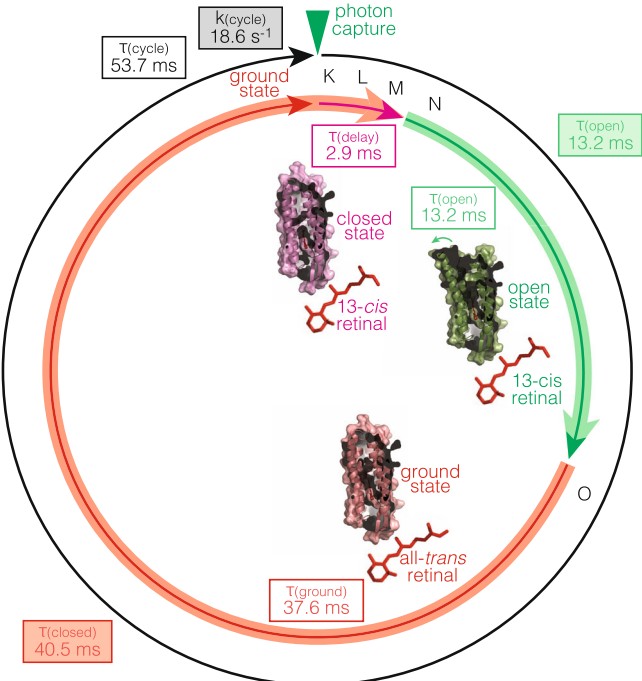

**Fig. 4 bR-WT reaction cycle kinetics in the purple membrane.** After photon capture and isomerization of the retinal from all-*trans* to 13-*cis*, bR is in a closed state structure for 2.9 ms (pink), comprising states K, L and M. For re-protonation of the retinal, bR opens the cytoplasmic gate displacing the E-F loop and stays in the open state structure for 13.2 ms (green), corresponding to state N. The green arrow highlights the movement of the E-F loop to open the cytoplasmic gate. After retinal re-protonation and isomerization to all-*trans*, bR relaxes to the dark state structure (red). Under continuous light at saturating conditions, the protomer stays in the closed state for 37.6 ms (40.5 ms − 2.9 ms), until it can be activated again. We propose that, during this time, the neighboring protomers cycle (2 × 2.9 ms + 13.2 ms = 32.2 ms). The structure shown are PDB 6RNJ, 6RPH and 6RQP[7].

back to the ground state, ~37.6 ms (40.5 ms closed state minus 2.9 ms opening delay time, end of N state to ground state). This is much longer than what spectroscopic measurements found, 1 ms to 10 ms, the decay time of those molecules in the ensemble that were in the given state[8,15]. In contrast to spectroscopic methods, our single molecule measurements capture precisely how fast an individual protomer in the purple membrane is able to perform, which crucially includes knowing how long it takes until that very molecule can reactivate, and the maximum turnover rate we found is ~20 s⁻¹, mainly limited by the long reset time. Arguably, AFM studies of membrane proteins imply that the sample is deposited on a support. However, in the physisorption process, no chemical or biological bond between sample and surface is formed, and a buffer layer is trapped between the membrane and the mica[48], maintaining the structural and functional properties of the purple membranes. Furthermore, no notable conformational changes (even at the atomic level) occur on the extracellular face, which faces the mica in our experimental setting when we investigate the cytoplasmic face. Thus, even if protein-support interactions would occur, no steric hindrance for conformational changes should be transferred to the cytoplasmic face of the protein. Also, the speed of the opening of the cytoplasmic gate (2.9 ms), corresponds well to the spectroscopic time constant, thus, it appears unlikely that a protein-support interaction would only concern the reverse conformational change. So how

can the surprisingly long inability to reactivate an individual protomer be explained? Both, spectroscopic and crystallographic methods may describe the decay of states, in this case of the O state, but cannot inform whether the very same protomer can reactivate. Interesting insight come however from crystallographic analysis: Indeed, the different state structures are calculated from difference maps from the ensemble diffraction data, and allow an estimate of the molecules in the open state. Such analyses indicated that ~30% of the molecules in these diffraction experiments were actually in the open state[7,42], and has been interpreted by Vonck[42] with steric hindrances in the native purple membrane for more than one protomer per trefoil (the interface between three trimers) to be in the F-helix extended conformation at any given time. Spectroscopic data, also indicated that only one molecule of three could be in the open state[49]. Such a model is in good agreement with our kinetic data. Indeed, our protomer reset time is ~37.6 ms, which would comprise well the activity of two neighboring protomers ~32.2 ms (2.9 ms opening delay time +13.2 ms open state dwell time = 16.1 ms × 2 neighboring protomers). Thus, we propose, that an isolated bR monomer could likely turnover at ~100 s⁻¹ but due to the native supramolecular structure in the purple membrane the protomer activity is limited to ~20 s⁻¹. Since our measurements are performed on isolated purple membrane sheets, H⁺-pumping does not build up an electrochemical gradient, and therefore the turnover rate of ~20 s⁻¹ at neutral pH and excess photon availability should represent the maximum rate. Future experiments on a protein interface mutant that is impaired to form 2D lattices could provide further insights into the influence of the 2D packing on the reaction cycle.

The physical distance ($d$) between the Lys216, where the retinal is attached and where the reaction cycle is initiated within picoseconds upon photon absorption, and Arg164 at the very top of the E-F loop is $d \approx 30\text{Å}$. The E-F loop moves $\tau_{delay} = 2.9$ ms later, thus the velocity of the allosteric coupling through bR is $v = d/\tau_{activation} = \sim 30\text{Å}/2.9ms \approx 1\mu m \cdot s^{-1}$. Another system for which we have a direct measure for the beginning and the endpoint of a conformational change are the voltage-gated ion channels: In voltage-gated ion channels, the gating currents report the movement of the voltage sensors and thus mark the starting point, while the ionic currents report the opening of the pore and therefore mark the end point of the conformational transition. The ionic currents occur ~1 ms after the gating currents, and the S4-S5 linker coupling distance is ~25 Å, giving a $v$ of $\sim 2.5\mu m \cdot s^{-1}$[50]. Thus, the allosteric coupling speed is of similar order as in bR, and we propose single digit $\mu m \cdot s^{-1}$ velocities as good estimates for conformational coupling velocity in membrane proteins.

At saturating photon availability, the single molecule reaction cycle in the purple membrane is $\tau_{cycle} = 53.7$ ms, and thus the maximum H⁺-pumping rate is $k_{H^+-pumping} = 1/(\tau_{cycle}) = 18.6 s^{-1}$. The purple membrane can be regarded as a minimal photosynthetic apparatus, where H⁺-pumping through bR fuels ATP-synthase action that generates the cellular energy currency ATP[51,52]. The ATP-synthase is constituted of two major parts, the F₀ rotor ring and F₁ where ATP formation takes place. F₀ functions in a turbine-like manner and is fueled through the flux of H⁺. It consists of about 10 subunits ($c_{ring} \approx 10$) and reaches a maximum rotation speed of $R_{ATPsynth} \approx 300s^{-1}$[53]. Thus $n \approx R_{ATPsynth} \cdot c_{ring}/k_{H^+-pumping} \approx 150$ bR protomers (50 trimers) would be sufficient to fuel one ATP-synthase under saturating light conditions.

In this work, we used HS-AFM-LS to analyze millisecond conformational dynamics in bR. The raw data provides single molecule traces of astonishing clarity at millisecond temporal resolution (Fig. 2a, Supplementary Movie 5). Given that the method is not limited to and influenced by the placement of labels and easily

reports Angstrom-range conformational dynamics at high temporal resolution[36,37], we propose HS-AFM-LS as a most powerful method to study membrane protein single molecule kinetics in general.

## Methods

**Sample preparation**. Purple membranes containing either bR-WT or bR-D96N mutant were isolated from *Halobacterium salinarum* as described previously[1]. A 2 μL drop of the purple membrane sample was deposited on a 1 mm$^2$ freshly cleaved mica surface. After incubation for 15 min, the sample was rinsed with imaging buffer (10 mM Tris-HCl (pH adjusted to the desirable value for each experiment), 300 mM KCl), and imaged with the HS-AFM.

**HS-AFM data acquisition and analysis**. All images were taken at room temperature with a sample scanning HS-AFM (RIBM, Japan) operated in amplitude modulation mode. Igor Pro version 6.37 was used for HS-AFM data collection. The cantilevers used were 8 μm long cantilevers with nominal spring constant k = 0.15 N/m and resonance frequency $f_0$ = ~600 kHz in buffer (1200 kHz nominal resonance frequency in air, USC-F1.2-k0.15, NanoWorld, Switzerland). Oxygen plasma etching was used to sharpen the tips. HS-AFM images were taken at 1-4 frames per second keeping the pixel-sampling ratio constant at 1.667 Å/px. Movie alignment was carried out in ImageJ with laboratory-build scripts.

HS-AFM-LS kymographs were recorded at 1.667 Å/px and a line scan rate of 1.667 ms. When switching from 2D imaging to HS-AFM line scanning, the central line of the former 2D scan is repeatedly scanned. Typically, a protomer is centered in the 2D scan before switching to HS-AFM-LS. However, owing to the dense packing of bR in the purple membranes, a high success rate for contouring protomers is achieved, wherever the line scan is taken. In brief, we recorded 600 left-right and right-left scan lines per second and each scan line has 300 pixels. Kymographs are composed of left-right scan lines only, each one acquired during 0.833 ms, however, each protomer is only probed once every 1.667 ms. We regard 1.667 ms the deadtime of our measurements: when the conformational change of a protomer is detected for the first time within in a pixel, it could have occurred right then or within the 1.667 ms before. Therefore, we set the activating laser pulse as zero-time mark, and bin events in 1.667 ms bins, e.g., 0–1.667 ms, 1.667–3.333 ms, etc.

For illumination of the sample, a green diode laser (wavelength: 520 nm) was used. The intensity measured after the objective lens was in the range 0.2–40 μW. To control laser light emission, a signal generator and a laser diode driver were used (NewPort 505B laser driver).

To analyze the displacement of the protomers in HS-AFM movies we used self-written routines in MATLAB (Matlab, Mathworks, USA) and in ImageJ. First, every monomer position within each frame was detected as the maxima above a given threshold. Second, for every monomer belonging to a trimer, the trimer center was determined and used as the origin position to measure the monomer displacement.

To analyze the kinetics of the protomers in HS-AFM-LS height-time traces, we used self-written routines in MATLAB and the Step Transition and State Identification (STaSi) method was used for the identification of the two states[36,43]. The STaSi algorithm uses hierarchical clustering to assign the optimum number of states.

To analyze the opening delay time, we recorded the photodiode signal along with the topography channel. From these two signals, we correlated exactly in which pixel the green laser started illuminating the sample. We defined the opening delay time as the time since the beginning of the pulse to the observation of the conformational change. Our line sampling rate is 1.667 ms therefore we are not able to record conformational changes of molecules within this period, and we bin all observations occurring within scan line cycles in bins, 0–1.667 ms, 1.667–3.333 ms, and so on (Supplementary Fig. 3). Thus, the opening delay times provided here give an upper limit.

**Laser light intensity**. To determine the irradiance, we measured the laser power above the fluid cell in the HS-AFM setup, just above the cantilever position. The photon flux was calculated as:

$$Irradiance\ or\ light\ intensity = \frac{Power}{A} \tag{1}$$

$$A = \pi r^2 \tag{2}$$

$$Photon\ Flux = \frac{Irradiance}{E_{ph}} = \frac{\lambda(520nm)}{h \cdot c} \cdot Irradiance \tag{3}$$

However, in order to determine the activating light intensity, we also need to determine the radius (r) of the laser beam spot at the sample position. This was achieved by covering the laser beam with a 50 μm wide cantilever. In this position, the laser beam is blocked by the wide cantilever (left part of trace). Next, we recorded the laser power while moving the cantilever through the light path in 1.6 μm steps, moving the cantilever completely out of the laser path. We found that the measured laser power increased over a cantilever displacement of ~15 μm (middle part of trace), until it plateaued again (right part of trace) (Supplementary Fig. 1). From this measurement we estimated that the laser beam radius (r) was

~7.5 μm. Together with the measurement of the power meter above the fluid cell, we could estimate the irradiance in our experiments.

**Reporting summary**. Further information on research design is available in the Nature Research Reporting Summary linked to this article.

## Data availability

All data supporting the findings of this study are available within the article and its supplementary information files. Data source are provided with this paper. Additional information and relevant raw data are available from the corresponding author upon reasonable request and at the earliest convenience. Source data are provided with this paper.

## Code availability

Data analysis was performed with customs scripts in Matlab R2018b and ImageJ FIJI v1.52. All codes are either published, ref. [36,37,40]. (https://github.com/George-R-Heath/AFM-LineScan-Analyser) and 43 (STaSi algorithm) or available from the corresponding author upon reasonable request and at the earliest convenience.

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

## Acknowledgements
The authors thank Claudiu Gradinaru, George Heath and Yining Jiang for assistance and discussions. This work was supported by grants from the National Institute of Health (NIH), National Center for Complementary and Integrative Health (NCCIH), DP1AT010874 (S.S.), National Institute of Neurological Disorders and Stroke (NINDS), R01NS110790 (S.S), and a Postgraduate grant in Life and Matter Sciences from the Fundación Ramón Areces (A.P.P.).

## Author contributions
A.P.P. and S.S. conceived and designed the experiments. A.P.P. performed the HS-AFM experiments. A.P.P. and A.M. built the HS-AFM light activation system. A.P.P. and S.S. analyzed the data. A.P.P. and S.S. wrote the manuscript.

## Competing interests
The authors declare no competing interests.
