## [Peer Review File · Nature Communications]

Single molecule kinetics of bacteriorhodopsin by HS-AFMREVIEWER COMMENTS

Reviewer #2 (Remarks to the Author):

The manuscript by Perrino et al employs high speed atomic force microscopy (HS-AFM) to study the kinetics of wildtype bacteriorhodopsin (BR) and its D96N mutant. The authors chose the D96N mutant because its slower dynamics allows full AFM movies to be captured. However, since wildtype BR undergoes significantly faster dynamics, the authors employed single HS-AFM line scans (LS) to obtain a higher temporal resolution (1.6 ms) for wt BR. HS-AFM-LS has been developed previously by the authors and applied last year to study dynamics of an amino acid transporter (Matin et al. Nature Commun.). Whereas in case of the transporter vertical domain movements were analyzed, for BR the light-induced lateral movement of the E-F surface loop was monitored. Although the dynamics of bacteriorhodopsin is well known from spectroscopy, NMR, and time-resolved crystallography, these methods only provide ensemble averaged data. The authors argue that their HS-AFM-LS method can detect E-F loop changes ("N state") in individual BR proteins under green laser light activation, thus achieving single molecule kinetics. The main result was that the same BR protomer can undergo maximally ~20 photocycles per second. Some secondary results include the speed of activation or dwells times at a range of pH values or light intensities for wildtype BR and B96N. Overall, the movies and data were quite nice and descriptive, but the writing could use a little improvement. A literature search revealed that most of the AFM results of BR D96N were also done by another group earlier (Shibata et al, Nature Nanotech, 2010), but the AFM kymographs of wt BR appear to be novel with respect to single molecule kinetics.

The general question arises whether there is a possibility that adsorption of purple membranes affects photocycle kinetics and causes the long reset time? The authors should discuss this point. Perhaps they performed some control experiments or have some ideas.

The authors also propose that an isolated BR monomer could have a faster turnover compared with trimers (100/s vs 20/s). Previous HS-AFM showed an effect of the purple membrane lattice on the rate of conformational changes, with lattice disruption slowing down the conformational change (Yamashita et al. 2013 J Struct Biol). How does this fit? Is the lattice in the BR HS-AFM-LS sample fully intact?

The following are suggestions to improve the manuscript:

(1) After reading the manuscript, I was left questioning whether or not AFM line scans provided reliable and interpretable information as the full AFM movies. It was not until I saw Supplementary Figure S2 that I was more convinced that particular line scans from the 2D AFM data (Fig S2B) could show reliable data. I think that Figure S2 should be discussed in the paper as a control experiment, demonstrating the link between the images, the line scans, and the kinetic data. Maybe even move Figure S2 to the main paper. Also, in the Methodology section, please discuss how the kymographs were chosen for wildtype BR (i.e. was it random or specific lines from an initial 2D scan?)

(2) Spell out the acronym, high-speed atomic force microscopy (HS-AFM) the first time it is used in the introduction.

(3) Figures are pretty good, maybe a little crowded, but some questions and suggestions are as follows:

- Figure 1e, f: The white dashed circle and yellow dashed circle are not overly distinguishable color choices for this figure. They could be changed.
- Figure 1d. What is the white “blob” in the middle of the movie/figure?
- Figure 4. Good summary figure, but perhaps the ground state structure should be moved closer to the word “ground state” on the cycle. Also, maybe label the EF loop or show an arrow denoting “opened” part of the structure.
- Figure 3. Define n (perhaps it’s the total number of analyzed BR molecules in one/multiple experiment?) Define (τ) as the exponential decay constant.

(4) It is probably better to replace “activated” and “inactivated state” nomenclature with “open” and “closed” state, respectively, throughout the entire text and figures. The authors used “activated” to refer to when the E-F loop is opened for the cytoplasmic gate (N-state), and used “inactivated” to refer to when the E-F loop is closed for the cytoplasmic gate (ground, J, K, L, M, O). It is a bit strange to use “inactivated” when including photo intermediates. Light-activated and nonactivated/resting would fit better.

(5) The ultrafast J state is mentioned in line 177, but the J state is not explained in the intro, other sections, or included in Figure 4. Change so that the document is self-consistent. Also, the list of states in line 176-177, and line 225 should be chronological. i.e. ground, K, L, M, and O... it’s weird that it is not.

(6) As a general point, the writing should be improved in spots, to remove long run-on sentences, provide clarification, and remove duplicates. Here are some examples.

- Line 28-30 – perhaps use light-induced intermediates of the reaction cycle. The justification in the brackets can perhaps move to the end of the sentence to take away less from the flow.
- Line 59-62 – run on sentence
- Line 172-178 – That’s a very long sentence. Please break up into at least two sentences.
- Line 342, line 344, remove “we know that”
- Line 349, line 367 - ~37.6ms (40.5ms inactivated state - 2.9ms activation time) is written word for word, twice.

- There are SO MANY “i.e.” in this work (13 by my count), and it takes away from the readability of the article. There are two in 389-390 alone. Try rephrasing.
- 188-192 Why are there two “:” in the same sentence??
- Line 267 – 272: There are two “i.e.”, a “:” and three sets of brackets in this sentence which may the sentence difficult to digest.
- Line 276-279: Awkward sentence.
- Page 5- Line 159-162 – Awkward/Disjointed placement of previously done experiments of glutamate transporter, separating information about BR.
- Every time “two states” is used, replace with “opened and closed state” for clarity.
- Line 384 – Are you measuring between the protonated base Schiff base and a particular residue? Can you specify?
- Line 43 – first ten of milliseconds is there a typo?

(7) Paragraph 395 - 400 – This paragraph is a bit awkward. It’s the first time that you mention c-ring and ATP-synthase action. A clarification sentence about this process would be useful to the audience.

Reviewer #3 (Remarks to the Author):

Perrino et al. explore the dynamics of bR using HS-AFM. They report several new properties for bR, including the gate opening time of 2.9ms after absorption and a period of 37.6ms “reset” time in which the complex is unable to open. The manuscript is concise and presents interesting single-molecule data about the cycle of bR when activated by green light. Authors also compare WT and D96N mutant. I think the manuscript provides an important piece to understanding bR, but also provides further proof for the applicability of another single-molecule modality of HS-AFM that can capture dynamics down to 1.7ms. I am therefore supportive of publication in Nat Comm; however, I have a few points that authors need to address:

- To accurately compare the WT and D96N mutant authors must use the same methodology to analyze the dynamics of the two systems. Although the dynamics of the D96N was captured by HS-AFM imaging, HS-AFM-LS should be used on the mutant as well to have an apples-to-apples comparison. Also, HS-

AFM-LS may reveal other possible phenomenon of the mutant, that may be hidden in the slower sampling method.

- Authors should discuss their findings for D96N mutant and compare with those of Ref. 28 (DOI: 10.1002/anie.201007544).

- The observation of two lifetimes at pH 5 and 6 (Table S1) for WT is discussed and speculation about potential mechanism is also provided, however the mutant is not analyzed in similar fashion. Even speculation of its behavior would strengthen and further improve the scientific value of the manuscript.

- Determination of the irradiation power is not completely accurate. At the very least authors should point out that the cantilever used during HS-AFM scanning is much smaller and thinner than the one used during the determination of the laser power. Metal layer thicknesses also vary between these two cantilevers and transmittance of gold varies significantly with thickness. E.g. at ~40nm thickness (coating of both sides on USC lever) transmittance of 500nm light should be around ~10%.

Minor issues:

- L281: ref missing.

REVIEWER COMMENTS

Reviewer #2 (Remarks to the Author):

The manuscript by Perrino et al employs high speed atomic force microscopy (HS-AFM) to study the kinetics of wildtype bacteriorhodopsin (BR) and its D96N mutant. The authors chose the D96N mutant because its slower dynamics allows full AFM movies to be captured. However, since wildtype BR undergoes significantly faster dynamics, the authors employed single HS-AFM line scans (LS) to obtain a higher temporal resolution (1.6 ms) for wt BR. HS-AFM-LS has been developed previously by the authors and applied last year to study dynamics of an amino acid transporter (Matin et al. Nature Commun.). Whereas in case of the transporter vertical domain movements were analyzed, for BR the light-induced lateral movement of the E-F surface loop was monitored. Although the dynamics of bacteriorhodopsin is well known from spectroscopy, NMR, and time-resolved crystallography, these methods only provide ensemble averaged data. The authors argue that their HS-AFM-LS method can detect E-F loop changes (“N state”) in individual BR proteins under green laser light activation, thus achieving single molecule kinetics. The main result was that the same BR protomer can undergo maximally ~20 photocycles per second. Some secondary results include the speed of activation or dwells times at a range of pH values or light intensities for wildtype BR and B96N. Overall, the movies and data were quite nice and descriptive, but the writing could use a little improvement. A literature search revealed that most of the AFM results of BR D96N were also done by another group earlier (Shibata et al, Nature Nanotech, 2010), but the AFM kymographs of wt BR appear to be novel with respect to single molecule kinetics.

Answer: We thank the reviewer for their overall positive assessment of our work. Indeed, our main achievement here is the single molecule conformational kinetics measurement of the wild-type bR with millisecond temporal resolution.

We agree with the referee that some of the work on D96N bR has been presented earlier in the papers the reviewer mentions. We cited all these works – references 26 to 28 in the manuscript – when presenting prior studies in the Introduction section. We have redone these measurements because we prefer to present a coherent story as an ensemble. In revision, we have calculated LAFM maps of the D96N mutant in the closed and open states, which allowed us to determine a 5.5Å E-F loop displacement towards the periphery of the trimer accompanied by a 9° counter-clockwise shift. Finally, the presentation of calculated kymographs from the slow imaging data allowed us nicely to explain what type of signal one would expect to see in the line scanning kymographs (as this reviewer points out in the specific point 1).

The general question arises whether there is a possibility that adsorption of purple membranes affects photocycle kinetics and causes the long reset time? The authors should discuss this point. Perhaps they performed some control experiments or have some ideas.

Answer: The reviewer is concerned that the adsorption of purple membranes on the mica sample support might impair the speed of the bR conformational cycle. Indeed, it is a general concern with AFM studies on membrane proteins that the sample must be deposited on a support. However, the physisorption process where no chemical or biological bond between sample and surface is formed, implies that a buffer layer is trapped between the supported lipid bilayer and the mica (Muller BJ 1997, ref 48 in the revised manuscript), maintaining the structural and functional properties of the purple membranes. It is also notable that even at the atomic level (X-ray structures) no significant conformational changes occur on the extracellular face, which faces the mica in our experimental setting when we investigate the cytoplasmic face. Thus, even if such protein-support interactions would occur, there should be no steric hindrance for conformational changes on the intracellular face of the protein. Also, the speed of the opening of the cytoplasmic gate (~2.9ms), M to N states, corresponds well to the spectroscopic time

constant, thus, it appears unlikely that a protein-support interaction would only concern the reverse conformational change. We discuss this topic in revision in lines (367-377).

Only indirectly related to the reviewer's question: For example, former spectroscopic measurements, Ultrafast infrared spectroscopy of bacteriorhodopsin (Diller BJ 1991), were performed on samples prepared in the following way "The sample is a thin film of purple membrane between two calcium fluoride windows... A purple membrane suspension in distilled water was dried to minimize the amount of water and then rehydrated until the photocycle was reestablished." In such a situation the bR purple membrane is a surface adsorbed paste, and still it displayed fast photocycle. While in such works only spectroscopic measurements were performed, it is reasonable to assume that if the photocycle remains preserved under such a harsh treatment, then the conformational cycle should be fine too in our much more physiological setting. bR is a very sturdy protein.

The authors also propose that an isolated BR monomer could have a faster turnover compared with trimers (100/s vs 20/s). Previous HS-AFM showed an effect of the purple membrane lattice on the rate of conformational changes, with lattice disruption slowing down the conformational change (Yamashita et al. 2013 J Struct Biol). How does this fit? Is the lattice in the BR HS-AFM-LS sample fully intact?

Answer: We think the reviewer is referring to our sentence: "Thus, we propose, that an isolated bR monomer could likely turnover at $\sim 100\text{s}^{-1}$ but due to the native supramolecular structure in the purple membrane the protomer activity is limited to $\sim 20\text{s}^{-1}$." Our statement relates to the spectroscopic measurements made on wild-type bR (Heberle PNAS 1992) where pH sensitive labels on the protein surface and in bulk were used to monitor the H^+ release and reuptake. In this work, based on the decay of the pyranine $\Delta\Delta A$ signal it was concluded that bR could reset after $\sim 10\text{ms}$, and thus turnover is as fast as 100 s^{-1} . As we discuss in the paper, this measurement is agnostic to when the very same single molecule could activate again. In reference 27, Yamashita et al. 2013 J Struct Biol, a W12I mutant is used to disrupt the bR 2D-lattice (residue 12 is at the protein-protein interface), and the D96N mutation is used to be able to assess the dynamics in HS-AFM imaging mode. In addition, in this work the molecules had to be anchored with the substrate to avoid that they would diffuse away. It was found that the ratio of activated molecules of the anchored double-mutant under continuous light rose slower than the D96N mutant in 2D-lattice (anchored or not). These results concern the efficiency of activating molecules and not the activation time or reaction cycle speed of a molecule (and of a double (W12I/D96N) or single (D96N) mutant that is impaired in its ability to reprotonate). Therefore, we think it is not possible to compare our results on the bR wild-type with these results. We interpret our results with regard to reference 41, Vonck 2000 EMBO J, that the outwards swing of helices E-F during the activation is only allowed for one protomer of three neighboring trimers at any given time, and this is why the reactivation of an individual protomer cannot occur again because in average the other two protomers in a given trefoil will activate next. Anyway, the reviewer's suggestion to analyze the behavior of WT in a system that is not constraint by a 2D lattice is good and we discuss about possible future experiments along these lines in the Discussion (lines 396-397).

The following are suggestions to improve the manuscript:

(1) After reading the manuscript, I was left questioning whether or not AFM line scans provided reliable and interpretable information as the full AFM movies. It was not until I saw Supplementary Figure S2 that I was more convinced that particular line scans from the 2D AFM data (Fig S2B) could show reliable data. I think that Figure S2 should be discussed in the paper as a control experiment, demonstrating the link between the images, the line scans, and the kinetic data. Maybe even move Figure S2 to the main paper. Also, in the Methodology section, please discuss how the kymographs were chosen for wildtype BR (i.e. was it random or specific lines from an initial 2D scan?)

Answer: We regret that the line scanning process was not well explained until Figure S2. Indeed, we added Figure S2, because we considered that reading the kymographs might not be that intuitive for researchers that have not acquired themselves such experimental data.

a) Indeed, as the reviewer points out in their general comment, the fact that in bR a loop moves from one to another position provides a particularly strong signal when the scan line coincides with the directionality of the loop motion (see Figure 2a), and gives an anticorrelated pattern when comparing the location where the loop is located in the dark closed state vs where it is in the open state. We detail in lines (175-177).

b) We discuss further Figure S2 as a control to showcase the type of signal that line scans produce (lines 142-144), but we prefer to let S2 in the Supplementary data, as it does not really contribute novel/different insights.

c) We added further details to the Methods section (lines 452-455). In brief, when switching from 2D scanning to line scanning, the central line of the former 2D scan is repeatedly scanned. One can place a protomer to the 2D scan center, but thanks to the dense packing of bR, there is literally always a couple of protomers that are nicely contoured wherever the line scan is taken on the purple membrane.

(2) Spell out the acronym, high-speed atomic force microscopy (HS-AFM) the first time it is used in the introduction.

Answer: We corrected this, in the Abstract (line 13) and the first appearance in the main text of the revised manuscript (line 62).

(3) Figures are pretty good, maybe a little crowded, but some questions and suggestions are as follows:

- Figure 1e, f: The white dashed circle and yellow dashed circle are not overly distinguishable color choices for this figure. They could be changed.

Answer: In revision, we have increased the width of the dashed circles significantly (to 5 points width) to make sure that the two circles are now distinguishable.

- Figure 1d. What is the white ‘blob’ in the middle of the movie/figure?

Answer: We did not comment about this, because the assignment is rather tentative and certainly not the topic of this manuscript. Anyway, we think that the ‘blob’, actually a ring, is a c-ring of the *h. salinarum* ATPase (subunit K). In an earlier work, Casuso BJ 2010, we have already seen these rings next to bR in native purple membrane samples. At the time, we indeed performed mass spectrometry analysis of the purple membrane sample and found traces of ATP synthase c-ring as a contaminant. This makes of course sense: We can consider the purple membrane as a primitive photosynthetic system, where the proton gradient generated by bR is used to fuel ATP synthesis through a proton driven ATP synthase. The reviewer can find the mass spec analysis in:

(<https://www.cell.com/cms/10.1016/j.bpj.2010.07.028/attachment/7c16b428-3ec8-4b76-91d0-9cfc1cacc1c2/mmc3.pdf>). The fact that the ‘blob’ is ring shaped, obviously matches well such a tentative assignment.

However, because this is much too tentative and not the topic of this work, we prefer to abstain from commenting about the topic.

- Figure 4. Good summary figure, but perhaps the ground state structure should be moved closer to the word “ground state” on the cycle. Also, maybe label the EF loop or show an arrow denoting “opened” part of the structure.

Answer: We thank the reviewer for their suggestions. In revision, we changed the color of the label “ground state” to red. In revision, the label “ground state” is placed near the ground state structure and at

the end of the cycle to clarify for non-specialist readers that the ground state must be reached to accept a new photoactivation. Regarding the open structure conformation, we have added an arrow near the EF loop to clarify the motion of the cytoplasmic gate opening (however, these structures are just there to complete the schematic of the cycle, detailed structural views are shown in Figure 1a,b). We also amended the caption to describe the arrow and provide the PDB access numbers of the structures.

• Figure 3. Define n (perhaps it's the total number of analyzed BR molecules in one/multiple experiment?) Define (τ) as the exponential decay constant.

Answer: In revision, we defined n and the dwell time in the caption of figure 2 and 3. It reads now:

Figure 2: n is the total number of events analyzed. τ denotes dwell time as the fit of the single exponential fit $y(x) = Ce^{-x/\tau}$

Figure 3: n is the total number of events analyzed. τ denotes dwell time as fit of the single exponential fit $y(x) = Ce^{-x/\tau}$ for pH>7 and double exponential fit $y(x) = C_1e^{-x/\tau_1} + C_2e^{-x/\tau_2}$ for pH 5 and 6.

(4) It is probably better to replace “activated” and “inactivated state” nomenclature with “open” and “closed” state, respectively, throughout the entire text and figures. The authors used “activated” to refer to when the E-F loop is opened for the cytoplasmic gate (N-state), and used “inactivated” to refer to when the E-F loop is closed for the cytoplasmic gate (ground, J, K, L, M, O). It is a bit strange to use “inactivated” when including photo intermediates. Light-activated and nonactivated/resting would fit better.

Answer: We agree with the reviewer and replaced ‘activated’ and ‘inactivated’ states with ‘open’ and ‘closed’ states, respectively, throughout the manuscript, and ‘activation’ time with ‘opening delay’ time.

(5) The ultrafast J state is mentioned in line 177, but the J state is not explained in the intro, other sections, or included in Figure 4. Change so that the document is self-consistent. Also, the list of states in line 176-177, and line 225 should be chronological. i.e. ground, K, L, M, and O... it's weird that it is not.

Answer: We agree with the reviewer's suggestion and eliminated the mention of the J state in line 177 in the revised manuscript. However, we want to maintain our description of the closed state as the ensemble of O, ground, K, L and M states when describing the continuous light experiments, because in the continuous light experiments, the conformationally closed state starts at O and continues into ground, K, L, and M. Especially given that we provide a circular scheme in Figure 4, we believe that this is actually the more comprehensive description of the experiments.

(6) As a general point, the writing should be improved in spots, to remove long run-on sentences, provide clarification, and remove duplicates. Here are some examples.

Answer: We thank the reviewer for their suggestions and separated long sentences into two whenever possible.

• Line 28-30 – perhaps use light-induced intermediates of the reaction cycle. The justification in the brackets can perhaps move to the end of the sentence to take away less from the flow.

Answer: Lines 28-32 in the revised manuscript read now:

The time scales at which the different light-induced intermediates of the reaction cycle occur, span from femtoseconds to milliseconds and have been studied by spectroscopy and structural methods (2-6). We prefer to use the terminology “reaction cycle” that represents better the conformational and functional dynamics than the more spectroscopic term “photocycle”.

• Line 59-62 – run on sentence

Answer: Lines 59-63 in the revised manuscript read now:

The structural aspects of the bR reaction cycle have been studied by several techniques, some of them are also time-resolved methods: X-ray crystallography (4, 16-18), solid-state NMR (19-21), cryo-EM crystallography (5, 22-25), high speed atomic force microscopy (HS-AFM, 26-32) and, more recently, time-resolved serial femtosecond crystallography (TR-SFX) and X-ray free electron lasers crystallography (XFELs) (3, 7, 33).

- Line 172-178 – That’s a very long sentence. Please break up into at least two sentences.

Answer: Lines X-X in the revised manuscript read now:

From the idealized state assignment traces, we recovered the dwell-times of the bR-WT closed (Fig. 2d, left) and open (Fig. 2d, right) states, which were, at $1.9 \cdot 10^{18}$ photons/cm²·s, $\tau_{\text{closed}} = 107.8 \pm 0.8$ ms and $\tau_{\text{open}} = 18.9 \pm 0.3$ ms, respectively. The closed state comprises all states in which the E-F-loop is in the position as in the dark state, thus O, ground, K, L, and M states. The open state is defined as the state in which the E-F-loop is in the position opening the cytoplasmic gate, thus the N state.

- Line 342, line 344, remove “we know that”

Answer: We agree, and removed “we know that” and rephrased accordingly. Line 354.

- Line 349, line 367 - ~37.6ms (40.5ms inactivated state - 2.9ms activation time) is written word for word, twice.

Answer: We agree and removed the explanation when it occurs the second time. We hope that all readers remember where the 37.6ms come from.

- There are SO MANY “i.e.” in this work (13 by my count), and it takes away from the readability of the article. There are two in 389-390 alone. Try rephrasing.

Answer: Following the reviewer’s suggestion, we have removed most of the “i.e.” from the text. The total count is now 2.

- 188-192 Why are there two “:” in the same sentence??

Answer: We thank the reviewer for pointing out our mistake, we corrected the punctuation in the revised manuscript.

- Line 267 – 272: There are two “i.e.”, a “:” and three sets of brackets in this sentence which may the sentence difficult to digest.

Answer: We modified the sentence for clarity in the revised manuscript (lines 277-283), it reads now:

The main impact of the D96N mutation is the delay in the re-protonation step. Indeed, the kymographs upon pulsed activation revealed that the bR-WT open state dwell-time, corresponding to the duration until the E-F loop regains its initial position, had a time constant $\tau_{\text{open-WT}} = 14.2$ ms (Fig. 3c, left, Fig. 3f,g), very similar to $\langle \tau \rangle_{\text{open-WT}} = 13.2$ ms from the continuous light experiment (Fig. 2g). In contrast, the bR-D96N protomers remained open over extended kymograph acquisition (Fig. 3d, left; Fig. 3b, Supplementary Fig. S5).

- Line 276-279: Awkward sentence.

Answer: We amended lines 287-290 in the revised manuscript:

In these experiments, the rise of the N state occurred ~3.5 ms after flash. The pyranine pH-probe fluorescence decayed ~12 ms and the N state decayed ~14 ms after the laser flash. These results are in good agreement with $\tau_{\text{delay}} = 2.9$ ms and $\tau_{\text{open}} = 13.2$ ms, found here.

- Page 5- Line 159-162 – Awkward/Disjointed placement of previously done experiments of glutamate transporter, separating information about bR.

Answer: This sentence has two objectives:

First, we want to point out the versatility of HS-AFM-LS for detecting protein conformational changes at high temporal resolution, in general.

Second, in the prior work HS-AFM-LS recorded vertical protein displacements, in other words height changes. In contrast in the case of bR, HS-AFM-LS detects the lateral displacement of the E-F loop. This lateral motion allowed us to analyze the two anticorrelated traces (red and green in Figure 2b), where the loop is in the closed state vs where the loop is in the open state. This is for the first time introduced in the sentence right after. Since many researchers think of AFM as a technique that is particularly sensitive in the detection of height changes (as in GltPh), we thought the comparison was important at this location where we detect the lateral shift of a protrusion that has literally almost no height difference in the open and closed states.

- Every time “two states” is used, replace with “opened and closed state” for clarity.

Answer: We checked the entire manuscript and found that in the revised version, we only say “two states” once when we describe the STaSi methods for general state assignment.

- Line 384 – Are you measuring between the protonated base Schiff base and a particular residue? Can you specify?

Answer: We amended the sentence:

The physical distance (d) between the Lys216, where the retinal is attached and where the reaction cycle is initiated within picoseconds upon photon absorption, and Arg164 at the very top of the E-F loop is $d \approx 30 \text{ \AA}$. (line 408-410)

- Line 43 – first ten of milliseconds is there a typo?

Answer: Yes, thank you for noticing. The sentence reads now: “within the first ten milliseconds” (line 43)

- (7) Paragraph 395 - 400 – This paragraph is a bit awkward. It’s the first time that you mention c-ring and ATP-synthase action. A clarification sentence about this process would be useful to the audience.

Answer: We wanted to give the readers a perspective about the coupling of the proton pumping by bR to ATP formation via proton reflux through the ATP synthase. Line 422-426 in the revised manuscript:

The purple membrane can be regarded as a minimal photosynthetic apparatus, where H⁺-pumping through bR fuels ATP-synthase action that generates the cellular energy currency ATP (51,52). The ATP-synthase is constituted of two major parts, the F₀ rotor ring and F₁ where ATP formation takes place. F₀ functions in a turbine-like manner and is fueled through the flux of H⁺. It consists of about 10 subunits ($c_ring \approx 10$) and reaches a maximum rotation speed of $R_ATPsynth \approx 300 s^{-1}$ (50). Thus $n \approx (R_ATPsynth \cdot c_ring) / k_H^{+-pumping} \approx 150$ bR protomers (50 trimers) would be sufficient to fuel one ATP-synthase under saturating light conditions.

We thought that readers will appreciate this discussion, especially since our result indicates a maximum pumping rate of $\sim 20 s^{-1}$ (instead of $\sim 100 s^{-1}$ as thought before). The simple back-of-the-envelope calculation indicates that at a bR pumping rate of $\sim 20 s^{-1}$ the ATP synthases are still fully fueled given the abundance of bR in the purple membrane.

Reviewer #3 (Remarks to the Author):

Perrino et al. explore the dynamics of bR using HS-AFM. They report several new properties for bR, including the gate opening time of 2.9ms after absorption and a period of 37.6ms “reset” time in which the complex is unable to open. The manuscript is concise and presents interesting single-molecule data about the cycle of bR when activated by green light. Authors also compare WT and D96N mutant. I think the manuscript provides an important piece to understanding bR, but also provides further proof for the applicability of another single-molecule modality of HS-AFM that can capture dynamics down to 1.7ms. I am therefore supportive of publication in Nat Comm;

Answer: We thank the reviewer for their overall positive assessment of our work.

however, I have a few points that authors need to address:

- To accurately compare the WT and D96N mutant authors must use the same methodology to analyze the dynamics of the two systems. Although the dynamics of the D96N was captured by HS-AFM imaging, HS-AFM-LS should be used on the mutant as well to have an apples-to-apples comparison. Also, HS-AFM-LS may reveal other possible phenomenon of the mutant, that may be hidden in the slower sampling method.

Answer: We agree with the reviewer that there is a general interest to using precisely the same method, in this case HS-AFM-LS, to analyze the WT and D96N mutant systems. However, HS-AFM-LS allows us to see at a time only a few molecules (~8 protomers) in a scan line, while HS-AFM imaging allows the analysis of more than 30 protomers at a time, but at reduced temporal resolution. From the imaging experiments we know that the D96N mutant stays in the open state for ~10.5s after photoactivation. As the reviewer can imagine, it is thus extremely challenging to record kymographs where essentially no drift occurs over such extended periods of time. In revision, we provide a new Supplementary Figure 5, in which such kymographs are shown. But, we cannot derive large statistics for dwell-time distributions analysis from these experiments.

- Authors should discuss their findings for D96N mutant and compare with those of Ref. 28 (DOI: 10.1002/anie.201007544).

Answer: We agree with the reviewer. Our D96N activated/open state decay times are in good agreement with Ref. 26-28. In Ref. 26 the authors reported the following open state decay times: 6.7±0.10s (pH7), 25±0.25s (pH8), 48±0.59s (pH9), and in Ref.27 10.4 ± 0.10s (pH7.5). In Ref. 28 the authors explore the use of blue light to speed up the inactivation of the protein. Their reported value in that article is 39±0.27s (pH8). We have not analyzed the response of the mutant at pH8, but we report 10.5s at pH7, in good agreement with Refs 26 and 27. In revision, we discuss this in lines 140-141.

- The observation of two lifetimes at pH 5 and 6 (Table S1) for WT is discussed and speculation about potential mechanism is also provided, however the mutant is not analyzed in similar fashion. Even speculation of its behavior would strengthen and further improve the scientific value of the manuscript.

Answer: Our main goal was to observe the WT-bR conformational changes with high temporal resolution, to extend our ability to measure bR kinetics with HS-AFM from former studies (Refs 26-28) that have explored the response of the D96N mutant in much detail. We refer the reviewer to Proc. Natl. Acad. Sci. USA Vol. 86, pp. 9228-9232, December 1989, where the D96N mutant has been studied spectroscopically across a wide range of pH.

- Determination of the irradiation power is not completely accurate. At the very least authors should point out that the cantilever used during HS-AFM scanning is much smaller and thinner than the one used during the determination of the laser power. Metal layer thicknesses also vary between these two

cantilevers and transmittance of gold varies significantly with thickness. E.g. at ~40nm thickness (coating of both sides on USC lever) transmittance of 500nm light should be around ~10%.

Answer: We think that the reviewer misunderstands the experiment shown in Supplementary Figure S1.

In order to calculate the irradiance, we needed not only to measure the laser power, but also determine the diameter of the laser beam. This is done using equations 1 and 2 in the original text:

$$\text{Irradiance or light intensity} = \frac{\text{Power}}{A}, \text{ (eq.1)}$$

$$A = \pi r^2, \text{ (eq.2).}$$

The experiment shown in Supplementary Figure S1 is only used to determine the diameter of the laser beam. For this, we used a 50 μm wide cantilever that covered the laser beam completely as it is moved in 1.6μm steps while recording the laser power: The flat laser power measurement in the left part of the trace in S1 means that the laser is entirely covered. The important region of this plot is where an increase in laser power is recorded, over about 15μm displacement in the middle part of the trace in S1. Thus, to clarify, the only objective of this experiment - when using the wide cantilever - is to determine the laser beam diameter. On the other hand, to evaluate the Irradiance, we put a laser meter above the fluid cell to determine how much light arrives at the sample. Knowing the laser power and the approximate size of the beam allowed us to estimate the irradiance.

Minor issues:

- L281: ref missing.

Answer: We regret having omitted the reference and thank the reviewer for pointing this out. We included Ref. 7 at this location in the text.

REVIEWERS' COMMENTS

Reviewer #2 (Remarks to the Author):

In the revised version of the manuscript the authors addressed all points of critique satisfactorily.

Reviewer #3 (Remarks to the Author):

Authors have improved the manuscript to a satisfactory degree. I do not have further comments.

The reviewers did not have any further requests or comments.
We thank them for their initial comments that helped us improve the manuscript.

REVIEWERS' COMMENTS

Reviewer #2 (Remarks to the Author):

In the revised version of the manuscript the authors addressed all points of critique satisfactorily.

Reviewer #3 (Remarks to the Author):

Authors have improved the manuscript to a satisfactory degree. I do not have further comments.